# Action-Conditioned Transformers for Decentralized Multi-Agent World Models

## Abstract

Multi-agent reinforcement learning (MARL) has achieved strong results on large-scale decision making, yet most methods are model-free, limiting sample efficiency and stability under non-stationary teammates. Model-based reinforcement learning (MBRL) can reduce data usage, but planning and search scale poorly with joint action spaces. We adopt a world model approach to long-horizon coordination while avoiding expensive planning. We introduce MACT, a decentralized transformer world model with linear complexity in the number of agents. Each agent processes discretized observation–action tokens with a shared transformer, while a single cross-agent Perceiver step provides global context under centralized training and decentralized execution. MACT achieves long-horizon coordination by coupling the Perceiver-derived global context with an action-conditioned contrastive objective that predicts future latent spaces several steps ahead given the planned joint action window and binding team actions to their multi-step dynamics. It produces consistent long-horizon rollouts and stronger team-level coordination. Experiments on the StarCraft Multi-Agent Challenge (SMAC) show that MACT surpasses strong model-free baselines and prior world model variants on most tested maps, with pronounced gains on coordination-heavy scenarios.

## 1 Introduction

Model free multi-agent algorithms such as QMIX Rashid et al. (2020), QPLEX Wang et al. (2020), and MAPPO Yu et al. (2022) can achieve robust long term returns, but they do so at the cost of millions of environment interactions. Two structural factors drive this sample hunger: the exponential growth of the joint observation action space as team size increases Liu et al. (2024) and the non stationarity that emerges when each agent's data distribution changes in response to its teammates' evolving policies Gronauer & Diepold (2022). For example, consider a 'focus fire' movement on the StarCraft multi-agent Challenge Samvelyan et al. (2019) (SMAC) environments, where a group of units must be commanded to attack a single enemy to eliminate it faster. Success depends on understanding the delayed consequences of the team's joint actions. This is the type of long horizon reasoning that models trained on one step prediction have a hard time grasping. In single agent settings, model based reinforcement learning (MBRL) addresses similar issues by training a latent world model Ha & Schmidhuber (2018) (WM) that can be rolled forward in imagi-

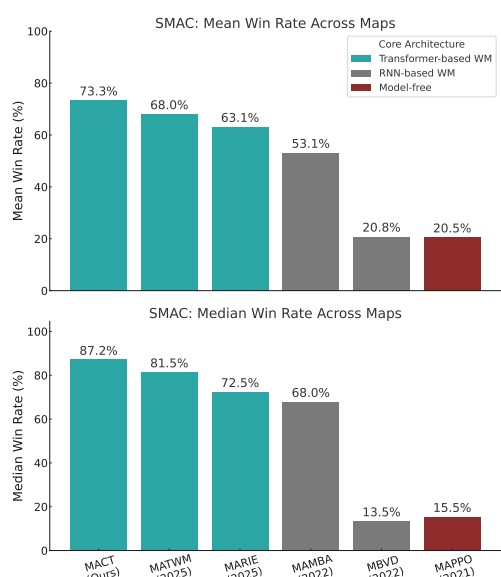

Figure 1: Mean (top) and median (bottom) win rates across SMAC maps. Bars are color-coded by their used learning methodology.

nation, thereby replacing expensive real transitions with synthetic ones, like Dreamer Hafner et al. (2019). The Dreamer transformer based successor TWISTER Burchi & Timofte (2025), and the

domain robust DreamerV3 Hafner et al. (2025) show that accurate latent dynamics can cut sample cost by an order of magnitude when the objective encourages multi step predictive structure.

Transferring this promise to multi-agent scenarios has proved difficult. MAMBA Egorov & Shpilman (2022) swapped Dreamer's recurrent core for an agent aware LSTM, yet all agents still shared the same latent state, so the model scaled poorly beyond a handful of entities. A more scalable design arrived with MARIE Zhang et al. (2025), which assigns each agent its own transformer for local token dynamics and injects joint context through a lightweight Perceiver cross attention layer. However, both MAMBA and MARIE supervise their models with one step token reconstruction losses, so the learned representations often capture only short term correlations and struggle whenever rewards depend on delayed or coordinated effects.

TWISTER showed in the single agent domain that a transformer equipped with action-conditioned contrastive predictive coding (AC-CPC) can exploit its full representational capacity. Instead of only reconstructing the next latent state, the model predicts a sequence of future latents $\{z_{t+1}, \ldots, z_{t+K}\}$ conditioned on the planned actions $\{a_t, \ldots, a_{t+K}\}$ and learns by contrasting the true future against negative samples in the batch. This long horizon objective rewards temporal abstractions that carry information several steps ahead and was critical for surpassing DreamerV3 on Atari 100k.

Our goal is to bring the benefits of model based learning to cooperative control without sacrificing simplicity or scale. We present **MACT**, a **M**ulti-agent **A**ction-**C**onditioned **T**ransformer that predicts several future latent states given a planned joint action sequence. Each agent is processed locally by a shared transformer, and a single Perceiver pass supplies light team context, which keeps computation near linear in team size. Figure 1 previews the results: across SMAC maps, world model approaches including MACT increase mean win rate under tight data budgets. In this paper we describe the objective and training procedure, evaluate on SMAC with ablations on horizon length and context, and analyze the coordination patterns that emerge.

Our contributions are described as follow:

- We designed an action-conditioned, multi-step contrastive learning objective for decentralized MARL: each agent predicts future Perceiver latents from its own short action window and current team context, with augmented positives to avoid trivial matching.

- On SMAC under tight data budgets, MACT yields higher mean and median win rates than strong model-free and prior world-model baselines, with the largest gains on coordination-heavy maps.

- Ablations show that moderate prediction horizons and light observation augmentation help, and that per-agent conditioning consistently outperforms team-aggregated conditioning.

## 2  RELATED WORK

Model-free multi-agent reinforcement learning (MARL) has progressed through value-factorization methods (VDN Sunehag et al. (2018), QMIX Rashid et al. (2020), QPLEX Wang et al. (2020)) and policy-gradient variants (MAPPO Yu et al. (2022), HAPPO Kuba et al. (2021)). All of them follow the centralized-training and decentralized-execution (CTDE) recipe: global information is used during learning, but each agent runs a local policy at test time. Because every joint configuration must still be sampled, their data budgets remain in the millions. This is an obstacle that our world model approach seeks to overcome.

Single-agent model-based RL pre-trains a generative model of environment dynamics and then improves a policy inside that model. Early versions such as SimPLe Kaiser et al. (2019) employed LSTMs, while Dreamer switched to a recurrent state-space model and introduced symlog rewards. DreamerV3 Hafner et al. (2025) refined the recipe, achieving domain robustness without per-task tuning. Several groups replaced RNNs with transformers to exploit parallel training: IRIS Micheli et al. (2022) maps each frame to a 4×4 grid of VQ-VAE van den Oord et al. (2017) tokens and processes the result with a spatial-temporal transformer; TWM Robine et al. (2023) concatenates observation, action, and reward tokens and trains a Transformer-XL; STORM Zhang et al. (2023) adds stochastic latent variables to a GPT-like backbone and reports strong human-normalized scores on Atari-100k. These works validate transformers as world model cores but still rely on next-step prediction and therefore do not fully tap long-horizon capacity.

Contrastive objectives address this limitation. CPC Oord et al. (2018) maximizes mutual information between present and future representations by contrasting the true future against negatives. In visual Reinforcement Learning (RL), the approach of using contrastive learning in combination with RL was popularized by the method CURL Laskin et al. (2020), which treats different data-augmented views of the same observation as a positive pair to learn spatial features but do not pay attention to the temporal and action-conditioned nature of control tasks. Building on temporal and action-driven features, TACO Zheng et al. (2023) introduces a temporal, action-driven contrastive loss designed to predict the future. TACO maximizes mutual information between a current state paired with a future action sequence and the resulting future state, which allows TACO to learn both state and action representations. A similar principle that applies in AC-CPC, which also includes the planned action sequence, removing ambiguities in passive video prediction. TWISTER Burchi & Timofte (2025) is the first to pair AC-CPC with a transformer world model, showing that long-horizon objectives unlock transformer capacity and surpass RNN baselines in low-data regimes.

Multi-agent world models face an additional scalability challenge: the joint observation–action space grows exponentially with team size. MAMBA Egorov & Shpilman (2022) adapts Dreamer to SMAC but keeps a single shared latent state, which limits scalability. MARIE Zhang et al. (2025) distributes token dynamics over agent-specific transformers and injects global context through one step of Perceiver cross-attention, achieving linear complexity in the number of agents. Other methods have explored augmenting the transformer world models with a teammate predictor module (MATWM) Deihim et al. (2025) or using learned models from value-decomposition data methods Xu et al. (2022). However, a common weakness in this prior works is the shallow one-step reconstruction loss, which causes rollouts to drift after a few steps, especially when rewards depend on coordinated actions spread over time.

## 3  METHODOLOGY

On SMAC environments, each map is modeled as a Dec-POMDP $\langle \mathcal{S}, \mathcal{A}^{1:N}, P, R, \Omega^{1:N}, \gamma \rangle$. At time $t$ every allied unit $i \in \{1:N\}$ receives a feature vector $o_t^i \in \mathbb{R}^{d_o}$ containing its own hit-points, cooldown, terrain height, relative distances to the nearest enemies and allies, and boolean flags. This boolean flags can be for example "enemy in range". The raw dimensionality is modest, around $d_o \approx 70$ for map 3s_vs_5z, but the joint observation space $\Omega^{1:N} = \Omega^1 \times \ldots \times \Omega^N$ still grows exponentially with $N$. The agent then chooses a discrete action $a_t^i \in \mathcal{A}^i$: *move-direction, attack-enemy, stop, and etc*. Executing the joint action $a_t^{1:N}$ through the unknown kernel $P$ yields the next state $s_{t+1}$ and the shared reward $r_t$. An episode ends when one army is eliminated or a time-limit is reached. The goal is to maximize the discounted return $\mathbb{E}\left[\sum_{t=0}^{\infty} \gamma^t r_t\right]$ while respecting decentralized execution. An overall view of our method is described in Figure 2.

*Vector-to-token conversion.* First, MACT converts each agent's continuous observation vector into a more structured sequence of discrete tokens. To do this, we use a small vector-quantized auto-encoder $(E, D, \mathcal{Z})$. This process creates a learned vocabulary for the features of the environment. The encoder takes the full observation vector $o_t^i$, splits it into $K = 8$ smaller pieces, and for each piece, it finds the best-matching "word", a code-book index $x_{t,j}^i \in \{1:256\}$, from its learned vocabulary. The result for this process is a compact sequence of $K$ tokens:

$$\boldsymbol{x}_t^i = (x_{t,1}^i, \ldots, x_{t,K}^i) \quad \text{with} \quad \boldsymbol{x}_t^i \in \mathcal{Z}^K. \tag{1}$$

Using discrete tokens instead of raw continuous numbers brings two practical advantages: it stabilizes training, because predicting the correct "word" from a fixed 256-entry vocabulary is a standard cross-entropy classification problem that avoids the large, unstable gradients of direct regression. At the same time, it exposes useful compositional structure, since treating observations as a token sequence lets the model learn language-like dependencies. One example is to make connections between a token for "small distance to an enemy" that will be often followed by "enemy in range". Finally, to form the input for a single time step, the $K$ observation tokens are concatenated with the agent's one-hot action $a_t^i$ and a placeholder aggregation token $*_t^i$ to yield the step block $\mathbf{X}_t^i = \left[\boldsymbol{x}_t^i, a_t^i, *_t^i\right]$.

*Local Transformer dynamics.* The transformer process an agent's history and produce a summary of its current situation. To prepare the input, token blocks are taken for each agent $i$ individually from the start of the episode up to the current time $0{:}t$ and flattened into a long sequence.

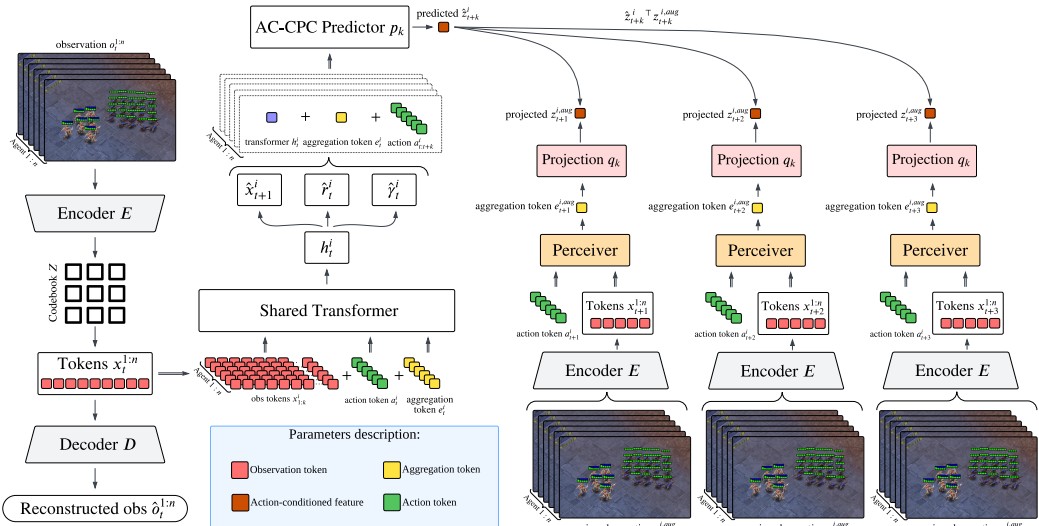

Figure 2: **MACT.** Tokenize observations with VQ, a shared Transformer plus a single Perceiver layer provides per-agent state $h_t^i$ and global context $e_t^i$. AC-CPC aligns $\hat{z}_{t+k}^i$ with projected Perceiver latents from a dropout-augmented future, given $[h_t^i; e_t^i; a_{t:t+k-1}^i]$ for $k=0{:}K_{\text{cpc}}-1$ with geometric weights. Default MACT's action-conditioning methodology is per-agent, in Section 4.1 we conducted an ablation study utilizing team aggregation.

A block-sparse Transformer $\phi$ processes this sequence. The block-sparse design is important: it restricts self-attention so that each agent's Transformer only focuses on its own history. In this stage, the transformer cannot see the raw history of other agents. But after reading its entire history, the model produces a single vector that summarizes the current step. This vector is the local summary, $h_t^i \in \mathbb{R}^{D_x}$. This summary is simply the Transformer's output state at the position of the aggregation token. Here $D_x$ is the token-embedding width (256).

*Centralized latent aggregation.* To inject team-level information we perform one Perceiver-style cross-attention update. We concatenate every agent's current tokens and embed them with $W_E$ to form

$$\mathbf{U}_t = \big[x_{t,1}^1, \ldots, x_{t,K}^1, a_t^1, \ldots, x_{t,1}^N, \ldots, x_{t,K}^N, a_t^N\big]W_E \in \mathbb{R}^{L \times D_x}, \qquad L = N(K+1). \quad (2)$$

We also maintain a learnable per-agent query matrix $\mathbf{Q} = [q^1; \ldots; q^N] \in \mathbb{R}^{N \times D_e}$ with $q^i \in \mathbb{R}^{D_e}$ ($D_e{=}256$). A single cross-attention layer then produces

$$(e_t^1, \ldots, e_t^N) = \mathbf{Q} + \text{softmax}\Big(\frac{\mathbf{Q}\mathbf{U}_t^{\top}}{\sqrt{D_e}}\Big)\mathbf{U}_t, \quad (3)$$

so each global latent $e_t^i$ mixes information from all agents. Finally, the $e_t^i$ vectors are appended as extra tokens to the next step's Transformer input, propagating global context forward.

*Prediction heads and one-step losses.* Following MARIE, the prediction heads attach directly to tokens produced by the local transformer. The observation head reads the $k$-th latent slot, not $e_t^i$, and outputs a categorical distribution over the code-book to predict $\hat{x}_{t+1,k}^i$ conditioned on $x_{\leq t,\cdot}^i$, $a_{\leq t}^i$, $e_{\leq t}^i$, and the previously generated slots $\hat{x}_{t+1,<k}^i$. This auto-regressive factorization across the $K$ slots captures intra-step structure such as the geometry of nearby units. The reward head maps the aggregation-slot hidden state $h_t^i$ through an MLP to produce a scalar reward prediction, $\hat{r}_t^i \sim p_\phi(\hat{r}_t^i \mid h_t^i)$. The discount head shares parameters with the reward head and predicts the Bernoulli continuation flag, $\hat{\gamma}_t^i \sim p_\phi(\hat{\gamma}_t^i \mid h_t^i)$. Finally, the one-step likelihood objective $\mathcal{L}_{\text{dyn}}$ is the sum of token cross-entropies, a SmoothL1 reward loss, and continuation binary cross-entropies over a replay segment of horizon $H$:

$$\mathcal{L}_{\text{dyn}} = \sum_{t=1}^{H}\big(\text{CE}(\hat{x}_{t+1,\cdot}^i, x_{t+1,\cdot}^i) + \text{SmoothL1}\big(\hat{r}_t^i, \text{symlog}(r_t)\big) + \text{BCE}(\hat{\gamma}_t^i, \gamma_t)\big). \quad (4)$$

*Action-conditioned contrastive prediction (per-agent).* Next-step supervision does not force the aggregation state $h_t^i$ to carry information about how *this agent's* planned actions will shape its future when teammates are also moving. Our AC–CPC objective therefore asks each agent to predict, in latent space, what its Perceiver context will be several steps ahead given its own action window. Concretely, for $k \in \{0 : K_{\text{cpc}} - 1\}$ we form the context

$$\underbrace{h_t^i}_{\text{Transformer}} \;\|\; \underbrace{e_t^i}_{\text{Perceiver}} \;\|\; \underbrace{a_{t:t+k-1}^i}_{\text{Agent actions}} \quad (\in \mathbb{R}^{D_x + D_e + k\,|\mathcal{A}|}), \tag{5}$$

where $h_t^i \in \mathbb{R}^{D_x}$ is the aggregation-slot hidden state of agent $i$ at time $t$, $e_t^i \in \mathbb{R}^{D_e}$ is its Perceiver-derived global context, and $a_{t:t+k-1}^i \in \{0,1\}^{k|\mathcal{A}|}$ is the concatenation of the next $k$ one-hot actions of agent $i$, while $|\mathcal{A}|$ is the per-agent discrete action vocabulary. A two-layer MLP $p_k : \mathbb{R}^{D_x + D_e + k|\mathcal{A}|} \to \mathbb{R}^{d_z}$ maps this concatenation to a projected prediction and a two-layer projector $q_k : \mathbb{R}^{D_e} \to \mathbb{R}^{d_z}$ produces the projected target,

$$\hat{z}_{t+k}^i = p_k\Big( \big[ h_t^i \,\|\, e_t^i \,\|\, a_{t:t+k-1}^i \big] \Big), \qquad z_{t+k}^i = q_k\big(e_{t+k}^i\big). \tag{5.1}$$

To avoid trivial token matching and encourage action-relevant invariance, the target $e_{t+k}^i$ is computed from an augmented observation view. With a minibatch of $Q$ positives (agent–time pairs), let $Z_{t+k} = [z_{t+k}^{(1)}, \dots, z_{t+k}^{(Q)}] \in \mathbb{R}^{d_z \times Q}$. The InfoNCE term uses dot-product logits $\hat{z}_{t+k}^{(q)\top} Z_{t+k}$ and cross-entropy over the index of the positive:

$$\ell_k = \tfrac{1}{Q} \sum_{q=1}^{Q} \text{CE}\big( \hat{z}_{t+k}^{(q)\top} Z_{t+k},\, q \big). \tag{5.2}$$

Intuitively, the predictor must learn a causal association: "if agent $i$ executes $a_{t:t+k-1}^i$ while embedded in team context $e_t^i$, its future context should look like $z_{t+k}^i$." This resolves temporal ambiguity (the action window disambiguates which future is correct), mitigates credit assignment by linking an agent's actions to its own future state, and reduces non-stationarity because teammates' reactions are already absorbed into $e_t^i$. We include a same-step term ($k{=}0$) to align spaces and stabilize training, and weight farther horizons geometrically with $\lambda_{\text{cpc}}{=}0.75$:

$$\mathcal{L}_{\text{cpc}} = \sum_{k=0}^{K_{\text{cpc}} - 1} \frac{\lambda_{\text{cpc}}^k}{\sum_{j=0}^{K_{\text{cpc}} - 1} \lambda_{\text{cpc}}^j}\, \ell_k, \tag{6}$$

ensuring gradients from distant steps remain present yet do not dominate.

*Available-action regularizer and total loss.* SMAC disables actions that are not possible through an "available-action" mask. Without extra care the logits of masked actions drift to large negative values, harming optimization. We add an auxiliary head that predicts next-step action availability. The head is trained with cross-entropy, and its contribution is added to the overall objective. Finally, the MACT loss is composed by the unweighted sum of token, reward, discount, availability, and AC-CPC terms:

$$\mathcal{L}_{\text{MACT}} = \mathcal{L}_{\text{token}} + \mathcal{L}_{\text{reward}} + \mathcal{L}_{\text{discount}} + \mathcal{L}_{\text{av-act}} + \mathcal{L}_{\text{cpc}}. \tag{7}$$

We found the natural scales of these terms to be comparable and therefore use equal weights. Training details and hyperparameters are deferred to Appendix B.

*Imagination-based actor–critic.* After every world model update we hold its weights fixed and unroll $H_{\text{roll}} = H$ latent steps. At each latent step every agent samples an action from a decentralized actor $\pi_\psi^i(h_t^i, e_t^i)$, the world model produces synthetic rewards and continuation flags, and rolls forward to the next latent. A central value network $V_\omega(e_t^1, \dots, e_t^N)$ evaluates global returns. Thus, $\lambda$-returns are back-propagated through the latent trajectory to train actors by advantage policy-gradient with an entropy bonus, and critics by symlog mean-squared error. Gradients do not flow through the frozen world model, ensuring stability.

## 4 EXPERIMENTS

We evaluate MACT on the StarCraft Multi-Agent Challenge (SMAC). The benchmark suite covers both easier micro scenarios (2m_vs_1z, 2s_vs_1sc, 3m, 8m, so_many_baneling) and

Table 1: Comparison of methods on SMAC. Values report median (std) win rate (%) over seeds.

| Map | Steps | MACT (Ours) | | MATWM Deihim et al. (2025) | | MARIE Zhang et al. (2025) | | MAMBA Egorov & Shpilman (2022) | | MBVD Xu et al. (2022) | | MAPPO Yu et al. (2022) | |
|---|---|---|---|---|---|---|---|---|---|---|---|---|---|
| | | Med | Std | Med | Std | Med | Std | Med | Std | Med | Std | Med | Std |
| 2m_vs_1z | 50K | 95.0 | 0.0 | **98.0** | 3.2 | 95.0 | 4.4 | 91.0 | 6.2 | 41.0 | 20.7 | 51.0 | 10.3 |
| 2s_vs_1sc | 50K | **98.7** | 2.2 | 96.0 | 5.7 | 90.0 | 9.1 | 80.0 | 7.3 | 0.0 | 1.2 | 18.0 | 7.6 |
| 2s3z | 50K | 65.0 | 1.7 | **80.0** | 9.0 | 71.0 | 8.6 | 68.0 | 12.1 | 28.0 | 17.5 | 13.0 | 3.0 |
| 3m | 50K | **96.7** | 3.4 | 83.0 | 10.4 | 78.0 | 14.1 | 68.0 | 7.7 | 60.0 | 9.2 | 54.0 | 6.3 |
| 3s_vs_3z | 50K | 80.0 | 3.4 | **87.0** | 19.4 | 85.0 | 21.8 | 77.0 | 23.7 | 0.0 | 0.0 | 0.0 | 0.0 |
| 3s_vs_4z | 50K | 4.1 | 7.2 | **12.0** | 4.8 | 0.0 | 0.8 | 4.0 | 1.4 | 0.0 | 0.0 | 0.0 | 0.0 |
| 8m | 50K | **95.0** | 5.0 | 67.0 | 24.9 | 72.0 | 7.1 | 68.0 | 6.4 | 52.0 | 18.9 | 38.0 | 4.9 |
| MMM | 50K | **39.2** | 32.6 | 7.0 | 4.7 | 1.0 | 1.6 | 3.0 | 3.5 | 0.0 | 0.0 | 0.0 | 0.0 |
| so_many_baneling | 50K | **94.4** | 7.9 | 86.0 | 22.9 | 73.0 | 12.4 | 66.0 | 14.2 | 27.0 | 12.3 | 31.0 | 7.6 |
| 3s_vs_5z | 200K | 65.0 | 11.7 | 64.0 | 26.5 | **66.0** | 28.0 | 6.0 | 10.1 | 0.0 | 0.0 | 0.0 | 0.0 |
| Mean | — | **73.3** | — | 68.0 | — | 63.1 | — | 53.1 | — | 20.8 | — | 20.5 | — |
| Median | — | **87.2** | — | 81.5 | — | 72.5 | — | 68.0 | — | 13.5 | — | 15.5 | — |

maps that demand tighter coordination or longer horizons (2s3z, 3s_vs_3z, 3s_vs_4z, MMM, 3s_vs_5z). Following the low–data protocol used by prior world model work Burchi & Timofte (2025); Zhang et al. (2025), we focus our main comparison on a $5 \times 10^4$ environment-frame budget. For a map that is particularly hard like 3s_vs_5z it was extend the training to $2 \times 10^5$ frames to allow for a clear performance difference. During training, we evaluate every 1,000 steps by computing the median win rate over 30 episodes. The final results are averaged across 3-4 random seeds. For instance, we use observation-dropout augmentation with $p$=0.03. The action-conditioned CPC horizon is $K_{cpc}$=8 on all maps except so_many_baneling and 2s3z, where $K_{cpc}$=5. Baselines include MAPPO Yu et al. (2022), MAMBA Egorov & Shpilman (2022) and MARIE Zhang et al. (2025), MATWM Deihim et al. (2025), and an MBVD variant Xu et al. (2022). Table 1 summarizes end performance, and Appendix A shows the learning curves.

Across tasks, MACT attains the best average performance (Mean row of Table 1: 73.3% vs. 68.0% for MATWM and 63.1% for MARIE) and ranks first on five of the ten maps. Gains are largest on coordination-heavy settings: on 8m MACT reaches 95.0% median win rate, on so_many_baneling 94.4%, and on MMM 39.2%. Relative to MARIE, the improvements are +23.0 points on 8m and +21.4 on so_many_baneling; relative to MATWM, the improvement on MMM is +32.2. MACT also converges rapidly on easier maps, achieving 98.7% on 2s_vs_1sc and 96.7% on 3m. Differences are small where performance saturates, as in 2m_vs_1z (95.0% vs. 98.0% for MATWM). On the 200k-frame 3s_vs_5z evaluation, MACT is comparable to MARIE (65.0% vs. 66.0%).

The learning curves indicate faster rise within the 50k-frame budget on the easier maps and larger seed variance on MMM, which is expected given the heterogeneous-unit interactions. MACT trails MATWM on 2s3z, 3s_vs_3z, and 3s_vs_4z (by 15.0, 7.0, and 7.9 points, respectively). These scenarios emphasize short tactical bursts and precise per-agent micro. A single Perceiver aggregation step may propagate less fine-grained context than required, and the shorter $K_{cpc}$=5 for 2s3z may contribute. Overall, the results support our central claim: coupling a Perceiver's global context with action-conditioned, multi-step prediction improves coordination under tight data budgets. Detailed analyses are presented in the following ablation studies.

## 4.1 ABLATION STUDIES

We conduct ablation studies to isolate the impact of MACT's key components. We chose to perform these studies on the 3m and 8m maps, as these are representative scenarios where our model achieved great performance. All ablations keep the architecture, optimizer, and implementation protocol fixed to Table 2. For all the conducted experiments, only the factor under test is varied. Panels (a)–(c) in Figure 3 correspond to the three experiments described in this section.

**CPC horizon.** We vary the action-conditioned CPC horizon $K_{cpc} \in \{4, 8, 12\}$ on 3m (Figure 3a). The $K$=8 model rises quickly and reaches the highest plateau, indicating that an 8-step action window is sufficient to cover the reaction–recovery cycle needed for coordinated focus fire. $K$=4 learns fastest at the very beginning but plateaus lower, consistent with under-specifying longer exchanges. $K$=12 eventually approaches the $K$=8 ceiling but starts noticeably slower and exhibits wider variability, reflecting heavier predictors and longer input action windows per InfoNCE term. Overall, an intermediate horizon balances temporal coverage and optimization stability.

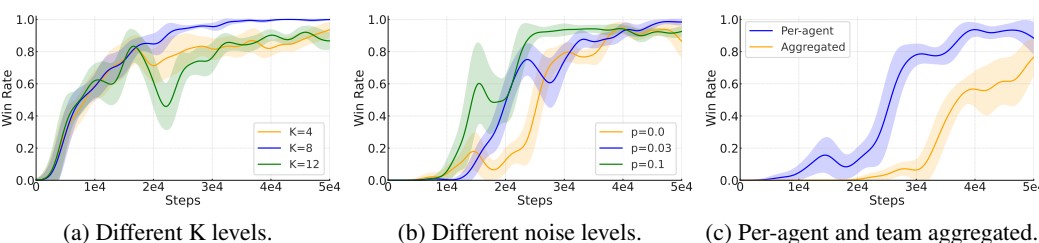

(a) Different K levels.   (b) Different noise levels.   (c) Per-agent and team aggregated.

Figure 3: Ablations study conducted on 3m and 8m SMAC environments.

**Observation dropout.** We tested how adding a small amount of noise, by feature-dropout, to the observations affects learning (Figure 3b). We tried feature-dropout probability $p \in \{0.0, 0.03, 0.1\}$ applied only when forming CPC positives on 8m. Without augmentation ($p$=0), learning onsets later and saturates lower, suggesting the contrastive task can be solved by brittle token matching. A small corruption ($p$=0.03) improves both speed and final win rate, pushing the predictor toward action-relevant invariance tied to Perceiver context rather than exact feature memorization. A heavier masking ($p$=0.1) can accelerate mid-training on some seeds but introduces oscillations and a slight late-stage drop, indicating that excessive noise erodes import information. A small, targeted amount of augmentation provides the best trade-off, encouraging the model to learn more robust features.

**Per-agent vs. team aggregated conditioning.** For completeness we test an agent-agnostic variant that replaces own actions with an aggregated joint-action summary at each step:

$$\tilde{a}^i_{t:t+k-1} = \mathrm{Agg}(a^{1:N}_{t:t+k-1}), \quad \mathrm{Agg} \in \{\text{mean}, \text{sum}, \text{max}\}.$$

To keep inputs comparable, the query features are also aggregated to team level ($h^{\text{team}}_t = \frac{1}{N}\sum_i h^i_t$, $e^{\text{team}}_t = \mathrm{Agg}(e^{1:N}_t)$) before applying the same Eq. (5)–(6).

On 8m, conditioning the CPC head on each agent's own future actions clearly outperforms using an aggregated joint-action summary (Figure 3c). The per-agent curve lifts off roughly $1 \times 10^4$ steps earlier, climbs steadily through mid-training, and reaches $\approx 0.9$ median win rate by $5 \times 10^4$ steps with tighter seed bands. The aggregated variant lags by about $1 - 1.5 \times 10^4$ steps and plateaus lower ($\approx 0.75 - 0.8$) with higher variance. We attribute this gap to identity-agnostic pooling blurring who executed which maneuver: AC-CPC learns best from a precise mapping between an agent's action window and the evolution of its Perceiver latent, while global team context is already provided by $e_t$. Adding a coarse joint-action summary appears redundant and noisier, increasing predictor input dimensionality and weakening contrastive alignment. We therefore adopt per-agent conditioning as the default conditioning methodology in MACT.

## 5 LIMITATIONS

This work offers an initial look at MACT rather than a definitive account. We focused our evaluation on a subset of SMAC with an emphasis on medium difficulty maps, extending the budget only for one harder scenario. A better picture, especially of scalability under very hard settings will require broader testing across the SMAC suite and additional cooperative benchmarks. Our experiments also use structured, vector observations, whereas the action-conditioned CPC objective that inspired our design was originally validated in visual single agent Atari. Understanding how MACT behaves with pixel inputs, where representation learning, tokenization, and optimization dynamics differ, is a natural next step (for example, visual SMAC variants or multi agent visual control tasks). Finally, our ablations were run on 3m and 8m, where we observed the clearest gains. While this choice improves measurement sensitivity, the patterns we report for horizon length, observation dropout, and per agent versus aggregated conditioning may not transfer unchanged to maps with different coordination structure. We view these as opportunities for extension, and systematic tests on harder environments and visual-based benchmarks will help establish the limits of the approach.

## 6 CONCLUSION

MACT is a decentralized action-conditioned transformer world model that couples a shared per-agent transformer and a single Perceiver step with an AC-CPC objective to tie planned actions to multi-step latent dynamics. On SMAC, it outperforms strong model-free and world model baselines, with the largest gains on coordination-heavy maps. Ablations indicate that intermediate CPC horizons, light observation dropout, and per-agent action-conditioning are key to the improvement. MACT can be integrated into imagination-based MARL pipelines. By adding a CPC head and conditioning on short action windows, the model develops a richer long-horizon structure and achieves stronger team coordination.

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

## A    APPENDIX: TRAINING RESULTS

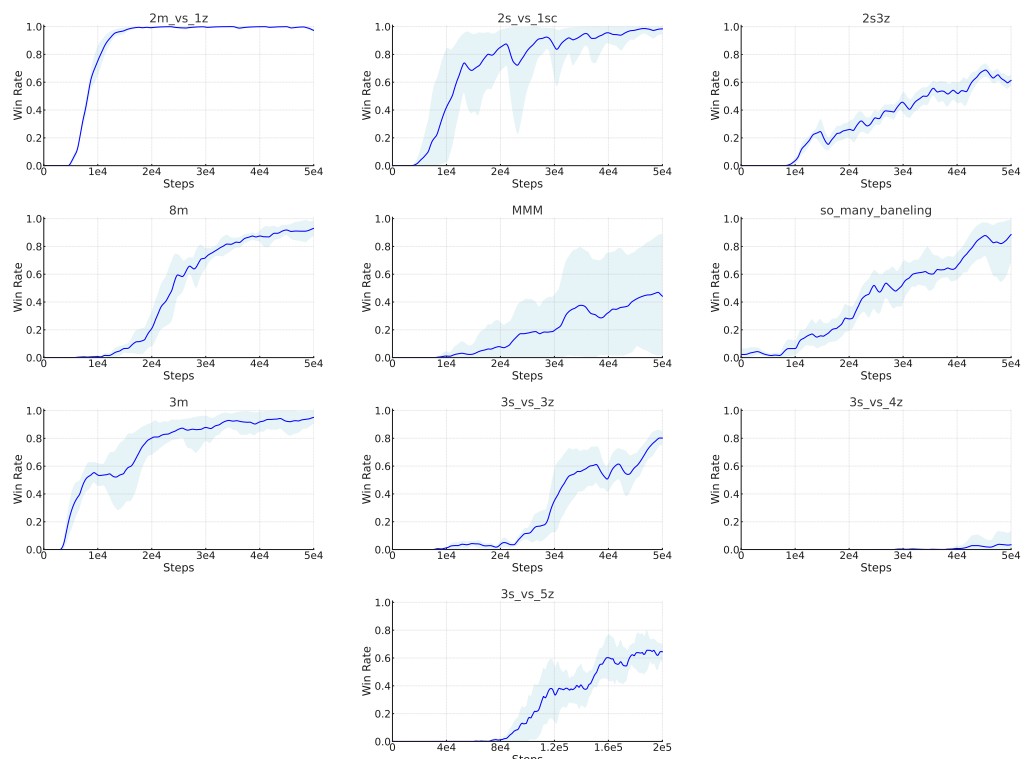

Figure 4: Evaluation win rate across all tested SMAC environments. During training, we ran 30 evaluation episodes every 1,000 environment steps.

## B    APPENDIX: MACT IMPLEMENTATION

**Reproducibility and setup.**    All ablations are run on the same implementation as our main method and keep MARIE's learning setup wherever applicable. To minimize environment drift, we pin the StarCraft II client to *SC2.4.1.2.60604* (the same version used by MATWM). It is worth to note that the use of different SC2 builds can yield slightly different win rates. Unlike MARIE (Python 3.7), our codebase targets Python 3.11 and PyTorch 2.x. Exact package versions, training scripts, per-map configs, the CSV files used to render plots, and the plotting scripts themselves are provided in the project repository[1] (see `requirements.txt` and the `data/` folders).

**Hyperparameters.**    We keep global training knobs identical across main and ablation runs (optimizer families, learning rates, clipping, entropy coefficient, etc.; see Table 2). As MARIE, for map-specific settings we select the imagination horizon $H$ and the *number of policy updates* as a function of the number of allied agents $N$: for $N \le 3$ (e.g., `2m_vs_1z`, `3m`, `3s_vs_5z`) we use $H=15$ and 4 updates; for $4 \le N \le 5$ (e.g., `2s3z`) we use $H=8$ and 10 updates; for $N \ge 6$ (e.g., `8m`, `MMM`, `so_many_baneling`) we use $H=5$ and 30 updates. Unless stated otherwise, the tokenizer and world model are each trained for 200 epochs, we perform 5 PPO epochs per policy update, and the $\lambda$-return uses $\lambda=0.95$ with $\gamma=0.99$. Map-specific overrides (e.g., $K_{\mathrm{cpc}}$) follow Table 2.

**What to toggle for each ablation.**    We change one switch at a time and keep everything else fixed. Action–conditioning is either per-agent or team (flag: `cpc_mode`); if team, choose how to aggregate joint actions (mean, sum, or max via `action_agg`). The CPC horizon and its geometric weighting use `K_cpc` and $\lambda_{\mathrm{cpc}}$ (defaults 8 and 0.75; map-specific overrides are in Table 2). InfoNCE

---

[1]Code available: `https://anonymous.4open.science/status/MACT`

Table 2: Hyperparameters for MACT in SMAC environments.

| Hyperparameter | Value |
|---|---|
| Batch size for tokenizer training | 256 |
| Batch size for world model training | 30 |
| Optimizer for tokenizer | AdamW |
| Optimizer for world model | AdamW |
| Optimizer for actor & critic | Adam |
| Tokenizer learning rate | 0.0003 |
| World model learning rate | 0.0003 |
| Actor learning rate | 0.0005 |
| Critic learning rate | 0.0005 |
| Gradient clipping (actor & critic) | max-norm 1 |
| Gradient clipping (tokenizer) | 10 |
| Gradient clipping (world model) | max-norm 1 |
| Weight decay for world model | 0.01 |
| $\lambda$ for $\lambda$-return | 0.95 |
| Discount factor $\gamma$ | 0.99 |
| Entropy coefficient | 0.001 |
| Buffer size (transitions) | $2.5 \times 10^5$ |
| Tokenizer training epochs | 200 |
| World model training epochs | 200 |
| Collected transitions between updates | $\{100, 200\}$ |
| Epochs per policy update (PPO epochs) | 5 |
| PPO clipping parameter $\epsilon$ | 0.2 |
| Number of imagined rollouts | 600 or 400 |
| Imagination horizon $H$ | $\{15, 8, 5\}$ |
| Number of policy updates | $\{4, 10, 30\}$ |
| Number of stacking observations | 5 |
| Observe agent id | False |
| Observe last action of itself | False |
| **AC-CPC** | |
| CPC horizon $K_{\text{cpc}}$ | $8^{\dagger}$ |
| Geometric decay $\lambda_{\text{cpc}}$ | 0.75 |
| Projection dimension (query/key) | 512 |
| Activation | ELU Clevert et al. (2016) |
| Action-conditioning | Per-agent action window $a_{t:t+k-1}^i$ |
| Negatives (InfoNCE) | In-batch |
| Positive target | Perceiver latent of augmented obs. |
| Vector-state augmentation | dropout $p$=0.03 |

Particularly, we used $^{\dagger}$ $K_{\text{cpc}}$=5 on `so_many_baneling`, `2c_vs_64zg` and `2s3z`; 8 on all other maps.

key detachment is a simple on/off (`detach_keys`). Reward targets use SmoothL1 on symlogged values by default. MARIE-style discrete two-hot is enabled via `use_ce_for_reward` (only used for `2m_vs_1z`). All ablations use the same evaluation protocol and plotting code.

**Pseudocode overview.** Algorithm 1 summarizes MACT in three phases. (i) *Experience collection* gathers short trajectories in SMAC and stores $\{o, a, r, \gamma\}$ in a replay buffer. (ii) *World–model update* samples a segment, builds per-step team context with one Perceiver cross-attention to obtain $e_t^{1:N}$, and runs the shared per-agent Transformer to produce $h_t^i$. The one-step likelihood (token, reward, discount, and optional availability heads) is combined with a *per-agent* action-conditioned CPC loss: for $k$=0:$K_{\text{cpc}}$−1, a predictor consumes $\left[h_t^i; e_t^i; a_{t:t+k-1}^i\right]$ (for $k$=0 the action window is empty) and is trained via InfoNCE against a *projected* Perceiver latent from an augmented future view; terms are weighted geometrically by $\lambda_{\text{cpc}}$. (iii) *Policy learning in imagination (CTDE)* freezes the world model and unrolls $H_{\text{roll}}$ latent steps; masked decentralized actors produce actions, a centralized critic evaluates returns, and $\lambda$-returns train the agents.

---

**Algorithm 1: MACT** (per-agent) training pseudocode

---

**Input** : Replay buffer $\mathcal{D}$;
horizons $H$ (model), $H_{\text{roll}}$ (imagination);
CPC horizon $K_{\text{cpc}}$;
decay $\lambda_{\text{cpc}}$
**Modules:** Tokenizer $(E, D)$;
shared per-agent Transformer $\phi$;
Perceiver cross-attn $\theta$;
CPC heads $\{p_k, q_k\}$;
decentralized actors $\{\pi_\psi^i\}$;
centralized critic $V_\xi$
**for** *epoch* $= 1, 2, \ldots$ **do**
    // (i) Collect real experience
    $o \leftarrow$ ENV.RESET(); **repeat**
        sample $a^i \sim \pi_\psi^i(\cdot \mid o^i)$ for $i{=}1{:}N$ (mask infeasible);
        $o', r, \text{done} \leftarrow$ ENV.STEP($a^{1:N}$);
        $\gamma \leftarrow \mathbb{I}[\neg\text{done}]$;
        push $\{o, a, r, \gamma\}$ to $\mathcal{D}$;
        $o \leftarrow$ done? ENV.RESET() : $o'$
    **until** *n steps*
    // (ii) Train world model
    sample segment $\{o_t^{1:N}, a_t^{1:N}, r_t, \gamma_t\}_{t=\tau}^{\tau+H-1} \sim \mathcal{D}$
    **Perceiver:**
    $e_t^{1:N} \leftarrow$ PERCEIVER$_\theta(E(o_t^{1:N}), a_t^{1:N})$
    **Transformer:**
    $h_t^i \leftarrow$ TRANSFORMER$_\phi([x_{\leq t}^i, a_{\leq t}^i, e_{\leq t}^i])$ for $i{=}1{:}N$
    **One-step loss:**
    $\mathcal{L}_{\text{dyn}} \leftarrow \mathcal{L}_{\text{token}} + \mathcal{L}_{\text{reward}} + \mathcal{L}_{\text{discount}} (+\mathcal{L}_{\text{avail}})$
    // Per-agent AC{CPC (InfoNCE with in-batch negatives)
    $o_t' \leftarrow$ AUGMENT_VECTOR($o_t$)           // dropout/noise on vectors
    $e_t'^{1:N} \leftarrow$ PERCEIVER$_\theta(E(o_t'), a_t)$         // augmented view
    **for** $k = 0$ to $K_{cpc}{-}1$ **do**
        **for** $i = 1$ to $N$ **do**
            $c_{t,k}^i \leftarrow [h_t^i; e_t^i; a_{t:t+k-1}^i]$     // empty action window if $k{=}0$
            $\hat{z}_{t+k}^i \leftarrow p_k(c_{t,k}^i)$          // query projection
            $z_{t+k}^i \leftarrow q_k(e_{t+k}'^i)$     // target projection (augmented future)
        stack $Z_{t+k} \leftarrow [z_{t+k}^{(q)}]_{q=1}^Q$ over all time–agent pairs in the minibatch
        $\ell_k \leftarrow \frac{1}{Q} \sum_{q=1}^Q \text{CE}(\hat{z}_{t+k}^{(q)\top} Z_{t+k}, q)$     // InfoNCE
    $\mathcal{L}_{\text{cpc}} \leftarrow \sum_{k=0}^{K_{\text{cpc}}-1} \dfrac{\lambda_{\text{cpc}}^k}{\sum_{j=0}^{K_{\text{cpc}}-1} \lambda_{\text{cpc}}^j} \ell_k$
    **Update** $(E, D, \phi, \theta, \{p_k, q_k\})$ by AdamW on $\mathcal{L}_{\text{WM}} = \mathcal{L}_{\text{dyn}} + \mathcal{L}_{\text{cpc}}$
    // (iii) Train actors in imagination (CTDE)
    init $o_0 \sim \mathcal{D}$,
    $e_0^{1:N} \leftarrow$ PERCEIVER$_\theta(E(o_0), a_0)$;
    **for** $t = 0$ to $H_{roll} - 1$ **do**
        sample $a_t^i \sim \pi_\psi^i(h_t^i, e_t^i)$ (mask infeasible);
        world model rolls to $(x_{t+1}^i, e_{t+1}^{1:N})$ and predicts $\hat{r}_t, \hat{\gamma}_t$
    compute $\lambda$-returns from $\{\hat{r}_t, \hat{\gamma}_t\}$;
    update $\psi$ by advantage PG (entropy bonus) and $\xi$ by symlog MSE

---

## C    Appendix: Use of Large Language Models (LLMs)

In accordance with ICLR 2026 LLM policies, we used LLMs only for copy-editing and for drafting minor plotting/figure scripts. All ideas and results are the authors' own, and all LLM output was reviewed and any code verified before use.

