# OpenReview forum: "Action-Conditioned Transformers for Decentralized Multi-Agent World Models"
_ICLR.cc/2026/Conference — ICLR 2026 Conference Withdrawn Submission_

### Official Review · Reviewer_S5t8 · 2025-10-31

**Soundness:** 3
**Presentation:** 2
**Contribution:** 2
**Rating:** 4
**Confidence:** 4

**Summary:**

This paper proposes a model-based approach to address the sample inefficiency of existing MARL methods. The authors introduce MACT (Multi-agent Action-Conditioned Transformer), a decentralized transformer-based world model. MACT processes each agent's local information with a shared transformer, integrates global context via a single Perceiver (following the CTDE paradigm), and learns long-horizon coordination through an action-conditioned contrastive objective that predicts future latent states several steps ahead based on the agent's own planned actions. Experiments on the StarCraft Multi-Agent Challenge (SMAC) demonstrate that the proposed method outperforms strong model-based and model-free baselines in a low-data regime.

**Strengths:**

1, The proposed MACT achieves a mean win rate of 73.3% across 10 SMAC maps in the low-data (50k frames) setting, outperforming strong baselines including MATWM (68.0%), MARIE (63.1%), and MAPPO (20.5%) (Table 1). The performance gains are particularly pronounced on coordination-heavy maps like 8m and so_many_baneling, with win rates of 95.0% and 94.4%, respectively.

2, The paper clearly demonstrates the effectiveness of its core contribution: per-agent action-conditioning. A direct comparison against a team-aggregated conditioning variant shows that using each agent's own action plan is substantially superior in both learning speed and final performance (Figure 3c), strongly supporting the design of the proposed objective.
3, The credibility of the method is enhanced by detailed ablation studies on its key components. The analysis of the CPC prediction horizon (Figure 3a) and the effect of observation dropout (Figure 3b) effectively isolates the impact of these hyperparameters, showing that an intermediate horizon (K=8) and a small amount of noise (p=0.03) yield optimal performance.

**Weaknesses:**

1, The proposed methodology is a clever combination of existing work. The architecture, using decentralized transformers with a centralized Perceiver for context aggregation, is very similar to MARIE (Zhang et al., 2025). The key differentiator, the action-conditioned contrastive predictive coding (AC-CPC) objective, is directly inspired by the single-agent work TWISTER (Burchi & Timofte, 2025). While the successful application and integration of this idea into the MARL domain is a contribution, it is difficult to argue that it presents a fundamentally new paradigm or methodology.

2, While the overall mean win rate is high, this appears to be driven by strong performance on relatively easier maps. On the scenarios that the authors themselves identify as demanding "tighter coordination or longer horizons" (e.g., 2s3z, 3s_vs_3z, 3s_vs_4z, 3s_vs_5z), the method's performance is either worse than or merely comparable to baselines like MATWM and MARIE (lines 283, 303-305).

3, Several of the ablation studies lack significant insight and feel superficial. The experiments on the CPC prediction horizon (K levels, Figure 3a) and observation noise (noise levels, Figure 3b) are akin to simple hyperparameter sweeps. They demonstrate that a certain setting performs better but fail to provide a deeper understanding of why it improves the model's underlying mechanics or learned representations. These results are of a level of importance that would be more appropriate for an appendix. More impactful analyses, like the per-agent vs. team-aggregated study, are needed to substantiate the paper's core claims, but these are diluted by less meaningful experiments.

**Questions:**

1, Could you provide a deeper analysis for why MACT underperforms baselines on the maps requiring harder coordination (2s3z, 3s_vs_3z, 3s_vs_4z, etc.)? This is critical for understanding the method's limitations and re-evaluating the central claim of improving long-horizon coordination.

2, Could you restructure the methodology section (Section 3) to clearly delineate between (1) components directly adopted from prior work (MARIE, TWISTER) and (2) the novel contributions of this paper (e.g., the specific method of integration, modifications for MARL)? This would allow for a clearer assessment of the paper's novelty.

3, To directly test the paper's central motivation against the limitations of one-step prediction, could you provide two key experiments?
(a) An ablation study for the AC-CPC objective with a horizon of K_cpc=1. This would directly verify if multi-step prediction in the world model's objective is indeed superior to one-step.
(b) An analysis of performance as the imagination horizon H_roll varies, crucially including the H=1 case. This would test whether the benefits truly come from long-horizon rollouts during policy learning.

---

### Official Review · Reviewer_tToN · 2025-11-01

**Soundness:** 3
**Presentation:** 2
**Contribution:** 3
**Rating:** 6
**Confidence:** 2

**Summary:**

The authors in this paper propose MACT, a multi-agent framework extending transformer-based world models like TWISTER to multi-agent settings, combining a shared per-agent transformer with a Perceiver cross-attention for team context. It uses an action-conditioned contrastive objective to learn long-horizon, coordination-aware latent dynamics under centralized training and decentralized execution. On SMAC, MACT achieves superior coordination and sample efficiency over prior world-model baselines.

**Strengths:**

1) The paper is strongly motivated. MARL is a highly relevant research direction.
2) The idea is intuitive and interesting. The method shows improved performance in co-ordination heavy maps of SMAC.

**Weaknesses:**

1) The work seems to be an extension of TWISTER to multi-agent systems.
2) The work should be evaluated on benchmarks with more complex observation space and continuous action space.

**Questions:**

1) Can the authors provide results on other benchmarks such as Google Research Football, MA-MuJoCo, etc?
2) Can the authors report the computational cost of the proposed method compared to the prior approaches?

---

### Official Review · Reviewer_NZ5K · 2025-11-01

**Soundness:** 2
**Presentation:** 1
**Contribution:** 1
**Rating:** 2
**Confidence:** 4

**Summary:**

This paper proposes MACT, a novel approach that learns a world model on a per-agent basis for model-based multi-agent reinforcement learning (MARL). A key component of MACT is its use of AC-CPC to estimate long-horizon future per-agent observations. The authors posit that by estimating long-horizon observations, rather than just one-horizon predictions, their method improves the performance of model-based MARL. The experimental results demonstrate that MACT achieves a higher win rate on the SMAC benchmark compared to other model-based MARL methods.

**Strengths:**

- The paper introduces MACT, a method that demonstrates superior performance over existing state-of-the-art model-based MARL algorithms.

**Weaknesses:**

- **Novelty:** The proposed MACT method appears to be a relatively straightforward combination of MARIE [1] and AC-CPC [2]. The paper's novelty would be significantly strengthened if it more clearly articulated the specific challenges encountered when combining these two approaches and detailed the novel architectural or algorithmic solutions developed to overcome them, rather than presenting what seems like a direct integration.

- **Clarity and Presentation:** The paper is difficult to read and understand.
  - Figure 2, which is intended to illustrate the MACT architecture, is confusing. It does not clearly explain the composition of the shared transformer's output, nor how the next observation token ($x^i_{t+1}$), reward ($r^i_t$), and discount factor ($\gamma^i_t$) are derived from the output $h^i_t$. (Based on the figure alone, $h^i_t$ appears to be a simple concatenation of $x^i_{t+1}$, $r^i_t$, and $\gamma^i_t$).
  - As a point of comparison, the figures in the original MARIE [1] and AC-CPC [2] papers are quite clear. The authors are encouraged to revise their figures to achieve a similar level of clarity and ease of understanding.
  - Furthermore, the descriptions of MACT's components and loss functions rely heavily on mathematical notation (detailing inputs and outputs) with insufficient high-level explanation. The paper lacks a clear articulation of why specific input/output structures were chosen for their intended functions, and it fails to provide a cohesive overview of how all components function and interact within the complete architecture.

- **Experimental Evaluation:** While the results in Table 1 are promising, model-based MARL evaluation critically depends on sample efficiency. Presenting learning curves instead of (or in addition to) final performance tables would provide a much more informative comparison of sample efficiency.

- **Ablation Studies:** The paper appears to lack sufficient ablation studies. It would be beneficial to see:
  - A comparison of MACT with K=1 (one-step prediction) to MARIE. Does it perform similarly? If not, what specific components account for the difference?
  - An analysis of the accuracy of the multi-step observation estimations compared to the actual ground-truth observations.


[1] Y. Zhang, et al., “Decentralized Transformers with Centralized Aggregation are Sample-Efficient Multi-Agent World Models,” Transactions on Machine Learning Research, Apr. 2025.

[2] M. Burchi, and R. Timofte, “Learning Transformer-based World Models with Contrastive Predictive Coding,” ICLR 2025.

**Questions:**

- Beyond the combination of MARIE and AC-CPC, are there any additional novel contributions or specific problems addressed and solved in this paper?

- Could the authors provide the main experimental results as learning curves to demonstrate sample efficiency?

- Are the ablation studies mentioned above (or other additional ablations) available?

---

### Official Review · Reviewer_YENi · 2025-11-04

**Soundness:** 2
**Presentation:** 1
**Contribution:** 2
**Rating:** 2
**Confidence:** 3

**Summary:**

This paper introduces MACT (Multi-Agent Action-Conditioned Transformer), a decentralized world model for multi-agent reinforcement learning that combines transformer-based architecture with action-conditioned contrastive predictive coding (AC-CPC). The method processes agent observations through a shared transformer with block-sparse attention, aggregates global context via a single Perceiver cross-attention layer, and trains using a multi-step contrastive objective that predicts future latent states conditioned on planned action sequences. Experiments on StarCraft Multi-Agent Challenge (SMAC) demonstrate improvements over model-free baselines and prior world model approaches, particularly on coordination-heavy scenarios.

**Strengths:**

1. **Well-motivated approach**: The paper clearly explains the limitations of one-step prediction objectives in multi-agent world models and provides strong motivation for extending AC-CPC from single-agent settings (TWISTER) to the multi-agent domain.

2. **Scalable architecture**: The combination of per-agent transformers with a single Perceiver aggregation step maintains linear complexity in the number of agents, addressing a key scalability challenge in multi-agent world models.

3. **Strong experimental results**: MACT achieves the highest mean (73.3%) and median (87.2%) win rates across SMAC maps, with particularly pronounced gains on coordination-heavy scenarios like 8m (+23.0 points over MARIE) and so_many_baneling (+21.4 points).

**Weaknesses:**

1. **The methodology section needs better organization**:
   - The logical structure of subsections is confusing. Subsections 3.1-3.3 describe the architecture, subsections 3.4-3.6 describe training objectives, and subsection 3.7 describes the training procedure. These are simply placed side by side without any integration or clear introduction.
   - It's difficult to connect the different modules and loss computations described in the text to the pipeline shown in the figure.
   - Does Figure 2 show only the world model training phase?
   - The paper uses a world model, policy, and critic, but doesn't describe the architectures of the actor and critic at all.

2. **Notation is confusing**. Some examples:
   - **W_E is undefined**: W_E appears in Equation 2 without prior introduction, and its role is unclear.
   - **Token representation confusion (Equations 1 vs 2)**: In Equation 1 (line 147), x^i_{t,j} ∈ {1:256} represents a discrete VQ code-book index after tokenization. However, in Equation 2 (line 196), the same notation x^i_{t,j} is multiplied by embedding matrix W_E, suggesting it's still a discrete index that needs embedding.
   - **Action representation confusion**: Line 158 explicitly states that a^i_t is a "one-hot action", but Equation 2's dimensionality L = N(K+1) treats each action as contributing a single token.

3. **MATWM lack of discussion**: MATWM (Deihim et al. 2025) appears to be closely related and is the most competitive baseline, achieving 68.0% mean win rate compared to MACT's 73.3% and outperforming MACT on 4 out of 10 maps. However, the authors only mention MATWM in a single sentence in the related work section (line 126) and briefly acknowledge it when discussing underperformance (lines 303-304).

4. **Limited experimental scope**: The evaluation focuses on medium-difficulty SMAC maps with vector observations under a 50K environment-step budget. Only one map (3s_vs_5z) extends the budget to 200K steps. The paper provides no evaluation on SMAC's harder scenarios (e.g., corridor, 6h_vs_8z) that would better test long-horizon reasoning claims, nor on any other multi-agent benchmarks beyond SMAC.

5. **Missing details on world model rollout budgets**: The paper specifies that MACT uses 600 or 400 imagined rollouts for policy training (Table 2), but doesn't state whether baselines (MARIE, MATWM, MAMBA) use the same number of rollouts.

6. **Citation formatting issues**: The paper has citation formatting inconsistencies throughout (e.g., inconsistent use of parenthetical versus textual citations, formatting of author names).

**Questions:**

see Weaknesses.

---

### Note · Authors · 2025-11-28

I have read and agree with the venue's withdrawal policy on behalf of myself and my co-authors.